# The increasing importance of fellowships and career development awards in the careers of early-stage biomedical academic researchers

Christopher L. Pickett ⬤ *

Rescuing Biomedical Research, Lewis-Sigler Institute, Princeton University, Princeton, NJ, United States of America

* clp3@princeton.edu

**Data Availability Statement:** All data used for these analyses were downloaded from the NIH ExPORTER public database (https://exporter.nih.gov/).

## Abstract

Excessive competition for biomedical faculty positions has ratcheted up the need to accumulate some mix of high-quality publications and prestigious grants to move from a training position to university faculty. How universities value each of these attributes when considering faculty candidates is critical for understanding what is needed to succeed as academic faculty. In this study, I analyzed publicly available NIH grant information to determine the grants first-time R01 (FTR01) awardees held during their training period. Increases in the percentage of the FTR01 population that held a training award demonstrate these awards are becoming a more common component of a faculty candidate's resume. The increase was largely due to an expansion of NIH K-series career development awards between 2000 and 2017. FTR01 awardees with a K01, K08, K23, or K99 award were overrepresented in a subset of institutions, whereas FTR01 awardees with F32 fellowships and those with no training award were evenly distributed across institutions. Finally, training awardees from the largest institutions were overrepresented in the faculty of the majority of institutions, echoing data from other fields where a select few institutions supply an overwhelming majority of the faculty for the rest of the field. These data give important insight into how trainees compete for NIH funding and faculty positions and how institutions prefer those with or without training awards.

## Introduction

Extreme competition for jobs, publications, and grant funding has been a defining feature of U.S. biomedical research for well over a decade, and this hypercompetition has outsized negative effects on early-career researchers [1–3]. The increased competition for jobs has put a spotlight on how graduate students and postdocs are trained for the wide variety of careers available to them [4]. Because of this focus on training, several structural changes in the graduate student and postdoc training regimen have been made, including the establishment of career development programs at a variety of institutions across the country [5]. More recently,

**Funding:** CLP is supported by a grant from the Open Philanthropy Project, https://www.openphilanthropy.org/. The funders had no role in study design, data collection and analysis, decision to publish, or preparation of the manuscript.

**Competing interests:** The author has declared that no competing interests exist.

new data collection initiatives have illuminated the variety of career paths trainees take once they move on from their graduate or postdoc programs [6–8].

The career path that we arguably have the most information about is the one that leads to an academic faculty career. Competition is fierce: nearly 80 percent of newly minted biomedical Ph.D.s enter a postdoctoral research position and more than half of these new postdocs intend to pursue a faculty position, despite this being far more postdocs than there are faculty slots [9–11]. Securing an F32 postdoctoral fellowship or a K-series mentored career development award from the National Institutes of Health can increase a candidate's chances of securing a faculty position and future funding, while having high-profile publications can be a predictor of success on the faculty track [12–14]. Postdocs also face a series of non-research related obstacles while trying to satisfy research requirements, including relatively low pay, decreased time for families and dimmed earnings prospects [11, 15].

However, our understanding of what makes a successful faculty candidate is still incomplete. For example, the NIH has nearly 30 different awards that trainees can apply for, and while receiving specific training awards confers an advantage in advancing along the faculty track, it is not known if all training awards are equally beneficial in this regard [12, 14, 16]. Graduate students, medical students, residents, and postdocs are eligible for NIH fellowships, designated as F-series awards (Table 1) [17]. The K-series NIH career development awards are directed to medical residents, postdocs, and some faculty (Table 1) [18]. As not all F and K-series awards have the same mission in supporting trainees, it is important to know which mechanisms predominantly support those who go onto successful faculty careers.

In the present study, I used publicly accessible information from the NIH to understand how important F and K awards were for those who received their first NIH R01 grant between 2000 and 2017. Researchers with faculty-level appointments are eligible to receive NIH

**Table 1. Information on select F and K-series awards.**

| Award | Name | Who can apply[a] | First awarded | Percent of category[b] |
|---|---|---|---|---|
| F awards | | | | 100 |
| F31 | Ruth L. Kirschstein Predoctoral Individual National Research Service Award | Graduate students | Late 1970s[c] | 15.8 |
| F32 | Ruth L. Kirschstein Postdoctoral Individual National Research Service Award | Postdocs/medical residents | Late 1970s[c] | 81.2 |
| Remaining F[d] | -- | -- | -- | 2.9 |
| K awards | | | | 100 |
| K01 | Mentored Research Scientist Career Development Award | Postdocs/early-stage faculty | 1968[e] | 18.9 |
| K08 | Mentored Clinical Scientist Research Career Development Award | Medical residents/early-stage faculty | 1974[e] | 35.5 |
| K23 | Mentored Patient-Oriented Research Career Development Award | Medical residents/early-stage faculty | 1999[e] | 18.5 |
| K99 | Pathway to Independence Award | Postdocs/medical residents/early-stage faculty | 2007 | 10.2 |
| Remaining K[f] | -- | -- | -- | 16.8 |

[a]Funding Opportunity Announcements restrict who can apply by, among other things, degree type and career stage.

[b]The number of awards of the indicated type divided by the total number of awards in the category in the pool of FTR01 awardees.

[c]See [17]

[d]Remaining F is made up of 7 F-series mechanisms. No remaining individual mechanism was over 1 percent of the category.

[e]See [18].

[f]Remaining K is made up of 19 K-series mechanisms. No remaining individual mechanism was over 5 percent of the category.

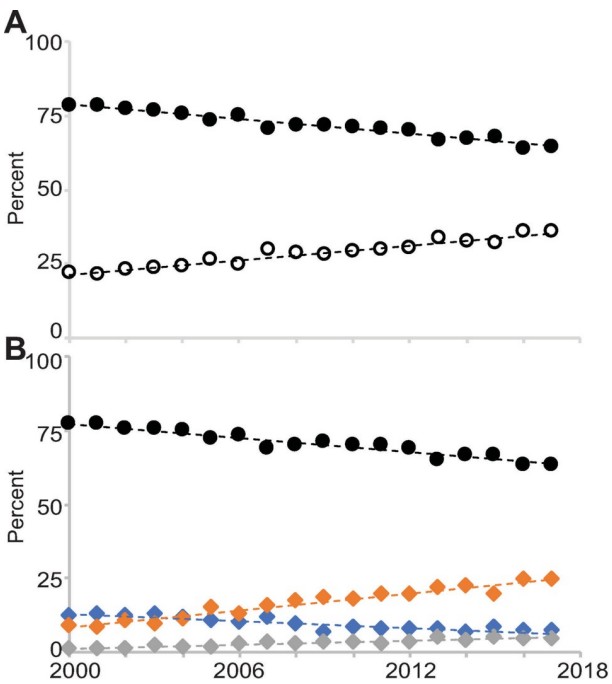

**Fig 1. The population of first-time R01 awardees with and without training awards between 2000 and 2017. (A)**
The percentage of the FTR01 population without a prior F or K award (closed symbol) and with a prior F and/or K
award (open symbol). **(B)** The percentage of the FTR01 population without a prior F or K award (black), those with an
F-series award (blue), K-series award (orange), and an F and a K-series award (gray). Best fit linear trend lines are
indicated as dashed lines.

research project grants, the most common and ubiquitous of which is the R01. R01s are highly
sought by early-career faculty and support more early-career faculty than any other NIH grant
mechanism [19, 20]. The data presented here indicate that a majority of first-time R01
(FTR01) awardees did not have an F or K award prior to attaining a faculty position, although
the percentage of the FTR01 awardee pool with an F or K award was increasing. I dissected
these data further to examine the five most common F and K awards and how these awardees
are distributed across the research enterprise. These data highlight differences in the distribu-
tion of FTR01 awardees with specific F and K awards, and I offer suggestions for further inves-
tigations to help shape policy.

## Results

I downloaded publicly available grant data from NIH ExPORTER to better understand the
relationship between F and K awards and subsequent R01s. Trainees are the principal investi-
gators on F and K-series awards, and they can be followed from their training position to their
faculty position with publicly available grant data. T-series training grants, on the other hand,
have a faculty member as the principal investigator, and the trainees funded by the T32 are not
in the public databases. For this study, "training awards" refers to F and K-series awards collec-
tively and does not include T-series training grants.

Using these data, I identified first-time R01 (FTR01) awardees in each year from 2000 to
2017. I then determined the F and K-series grants these FTR01 awardees held prior to their
R01 dating back to 1985 (See Data collection and limitations). In 2000, about 23 percent of the
FTR01 awardee pool had previously held a training award, and this rose near 37 percent in
2017 (Fig 1A and S1 Table). The percentage increased linearly across this time ($r^2$ = 0.95; Fig

1A). These data indicate that a minority of faculty have held an NIH training award prior to becoming faculty, but that holding a training award is becoming an increasingly common attribute of FTR01 awardees.

I broke the population of those with a training award into those who held an F-series award, a K-series award, and one of each. In 2000, 77 percent of FTR01 awardees had never held an NIH grant, 9 percent had held a K award, 12.5 percent had held an F award, and 1.5 percent had held an F and K award prior to receiving their first R01 (Fig 1B and S1 Table). By 2017, 63.3 percent of FTR01 awardees had never held an NIH grant (1.2-fold decrease over 2000; linear trend line $r^2 = 0.93$), 24.5 percent had held a K award (2.7-fold increase; $r^2 = 0.95$), 7.4 percent had held an F award (1.7-fold decrease; $r^2 = 0.82$), and 4.8 percent had held an F and a K award prior to receiving their first R01 (3.3-fold increase; $r^2 = 0.87$; Fig 1B and S1 Table). These data indicate the decline in the proportion of the FTR01 awardee pool that never held a training award arose almost entirely from an increase in those who held K awards.

F and K-series awards are a collection of individual grant mechanisms. To determine if changes in the abundance of these grants can account for the change in prevalence of F and K-series awards in the FTR01 awardee pool, I determined the most common F and K-series mechanisms in the FTR01 awardee pool. There were nine F-series mechanisms in the FTR01 awardee pool between 2000 and 2017. Almost 16 percent were F31 predoctoral fellowships and over 80 percent were F32 postdoctoral fellowships (Tables 1 and S2). No other F-series mechanism, including the F30 MD/PhD predoctoral award, accounted for more than one percent of F awards in the FTR01 pool (S2 Table). There were 23 K-series mechanisms in the FTR01 awardee pool in the same time frame, and K01, K08 and K23 mentored career development awards and the K99 pathway to independence award combined to account for 83 percent of K-series awards (Tables 1 and S2). No other K-series mechanism accounted for more than 5 percent of K awards in the FTR01 awardee pool (S2 Table). As the F32, K01, K08, K23, and K99 mechanisms were likely to be the training awards most proximal to receipt of an R01, I focused my remaining analyses on these five mechanisms.

I next assessed whether changes in the percentages of F32, K01, K08, K23, and K99 awardees in the FTR01 awardee pool could account for the overall F and K-series changes between 2000 and 2017 (Fig 1B). The percentage of FTR01 awardees with a prior F32 award declined from 2000 to 2017 by about a third (Fig 2A and S3 Table). The percentage of FTR01 awardees with K01, K23, or K99 awards each increased across the time frame, while those with a K08 were relatively stable (Fig 2A and S3 Table). These data indicate that the broad changes in F and K-series awards in the FTR01 awardee pool are mirrored by changes in these five individual mechanisms. It is important to note that the K01 program was a very small program until it was expanded in the late 1990s, K23 grants were first awarded in 1999, and K99 grants were first awarded in 2007 (Table 1) [18]. Therefore, the increase in the percentage of the FTR01 pool with these awards is to be expected. Furthermore, it appears that it took about 10 years for K01 and K23 awardees to become a stable percentage of the FTR01 population (Fig 2A and S3 Table). K99 awards have now been awarded for ten years, but it will take a few more years to determine if the number of K99 awardees in the FTR01 awardee pool has similarly stabilized.

The changes of these five training awards in the FTR01 awardee pool could be due to changing university preferences for hiring training awardees or simply by changes in the absolute number of training awards made over this period. The total number of F32 and K08 awards fell between 2000 and 2017, while K01 and K23 awards increased (Fig 2B and Table 2). K99 awards, first made in 2007, also increased between 2007 and 2017 (Fig 2B and Table 2). The trends for F32, K01, K23, and K99 awards match what might be expected from the percentage of FTR01 awardees with these awards (Fig 2B and Tables 2 and S3). The exception is

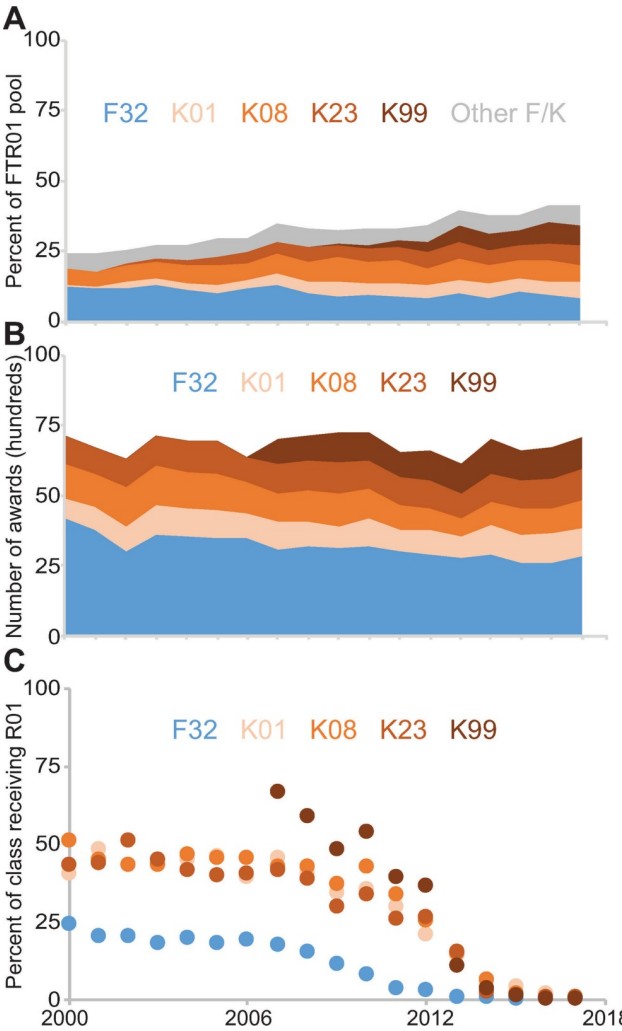

**Fig 2. Changes in indicated training awards between 2000 and 2017.** (**A**) Percent of FTR01 awardees with the indicated training award. (**B**) Number of indicated training awards made annually. (**C**) Percent of training awardees from each year to go on to receive an R01.

the K08 awards, which was flat in the FTR01 awardee pool while falling in real numbers (Fig 2B and Tables 2 and S3).

Changes in the absolute numbers of awards could reduce the number of FTR01 awardees with a specific training award, but there could be an additional effect if a university's hiring attitudes toward candidates with specific awards changed to over or undervalue them. Therefore, I next tested whether there was any change in the percentage of the training awardees from each year to eventually receive an R01. For those receiving an F32 between 2000 and 2007, roughly 20 percent each year went on to receive an R01 (Fig 2C and Table 2). For those receiving a K01, K08 or K23 between 2000 and 2010, around 40 percent each year eventually received an R01 (Fig 2C and Table 2). And for those receiving a K99 between 2007 and 2010, about 55 percent each year eventually received an R01 (Fig 2C and Table 2). The percentage of F32 awardees receiving an R01 declined after 2007 and the percentage of K01, K08, K23, and K99 awardees declined after 2010. This is likely because the awardees have not had sufficient time to complete their training, find faculty positions and successfully compete for an R01.

**Table 2. Percentage of training awardees who go on to receive an R01.**

| Year | F32 | | K01 | | K08 | | K23 | | K99 | |
|---|---|---|---|---|---|---|---|---|---|---|
| | Total awards[a] | Receiving R01 (%)[b] | Total awards[a] | Receiving R01 (%)[b] | Total awards[a] | Receiving R01 (%)[b] | Total awards[a] | Receiving R01 (%)[b] | Total awards[a] | Receiving R01 (%)[b] |
| 2000 | 836 | 24.4 | 141 | 40.4 | 256 | 50.8 | 193 | 43.0 | -- | -- |
| 2001 | 759 | 20.2 | 160 | 48.1 | 241 | 44.8 | 182 | 43.4 | -- | -- |
| 2002 | 599 | 20.4 | 178 | 43.3 | 290 | 43.1 | 196 | 51.0 | -- | -- |
| 2003 | 718 | 18.2 | 215 | 43.7 | 280 | 43.2 | 212 | 44.8 | -- | -- |
| 2004 | 711 | 19.5 | 198 | 44.9 | 264 | 46.6 | 223 | 41.3 | -- | -- |
| 2005 | 699 | 18.2 | 195 | 46.2 | 268 | 45.5 | 229 | 39.7 | -- | -- |
| 2006 | 692 | 19.1 | 177 | 39.0 | 228 | 45.6 | 178 | 40.4 | -- | -- |
| 2007 | 616 | 17.5 | 203 | 45.3 | 191 | 42.4 | 215 | 41.4 | 182 | 66.5 |
| 2008 | 638 | 15.2 | 176 | 40.3 | 228 | 42.5 | 213 | 38.5 | 178 | 59.0 |
| 2009 | 630 | 11.4 | 151 | 34.4 | 239 | 37.2 | 225 | 29.8 | 203 | 48.3 |
| 2010 | 642 | 7.8 | 192 | 35.4 | 215 | 42.3 | 208 | 33.7 | 191 | 53.9 |
| 2011 | 599 | 3.5 | 155 | 29.7 | 175 | 33.7 | 202 | 25.7 | 180 | 39.4 |
| 2012 | 584 | 2.9 | 168 | 20.8 | 163 | 25.2 | 200 | 26.5 | 211 | 36.5 |
| 2013 | 551 | 0.9 | 161 | 14.3 | 128 | 14.8 | 177 | 15.3 | 211 | 10.9 |
| 2014 | 585 | 0.5 | 202 | 5.9 | 168 | 6.5 | 200 | 2.5 | 245 | 3.3 |
| 2015 | 518 | 0 | 207 | 4.3 | 182 | 1.6 | 205 | 1.5 | 209 | 1.4 |
| 2016 | 517 | 0 | 212 | 1.9 | 183 | 0.5 | 205 | 0.5 | 230 | 0 |
| 2017 | 570 | 0 | 203 | 0 | 199 | 0 | 216 | 0.5 | 234 | 0.4 |
| % Change[c] | -26.3 | -- | 36.2 | -- | -16.0 | -- | 7.8 | -- | 4.9 | -- |
| Average[d] | -- | 19.7 | -- | 41.9 | -- | 44.0 | -- | 40.6 | -- | 56.9 |
| StDev[d] | -- | 2.1 | -- | 4.4 | -- | 3.4 | -- | 5.6 | -- | 7.7 |

[a]The total awards of the indicated type granted by the NIH in the indicated year.

[b]The percentage of those receiving the indicated training award in the indicated year who eventually receive an R01.

[c]The percent change in the total number of awards between 2000 and 2007 (F32), 2000 and 2010 (K01, K08, K23), and 2007 and 2010 (K99).

[d]The average and standard deviation of the percent of the pool receiving an R01 between 2000 and 2007 (F32), 2000 and 2010 (K01, K08, K23), and 2007 and 2010 (K99).

These data indicate the changes in the composition of the FTR01 awardee pool are likely driven by training award abundance.

## Distribution of training awards and FTR01s among institutions

Institutions evaluate faculty candidates on their funding track records, among other things. To gain a more granular understanding of the interplay between F32, K01, K08, K23, and K99 awards, subsequent R01 awards, and the roles of institutions, I divided institutions into quartiles based on the number of FTR01s they received from 2000 to 2017 (See Data collection and limitations). The FTR01s for institutions were summed across the years, and the institutions were arrayed from those that received the most FTR01s to the fewest. I then split the institutions evenly into quartiles based on the number of FTR01 awards received (S4 Table). The first quartile was comprised of 12 institutions, the second quartile of 23 institutions, the third quartile of 46 institutions, and the fourth quartile of the remaining 896 institutions (S4 Table).

I first determined whether FTR01 awardees differentially segregate into quartiles based on the type of training award received. Because the original list or universities was divided evenly into quartiles based on the number of FTR01s received, the prediction is that each quartile

should have about 25 percent of FTR01 awardees with a specific training award if there was no quartile-specific preference for hiring candidates with that award. Quartile values for FTR01 awardees with no prior award and those with a prior F32 were between 21 and 28 percent, indicating faculty with no prior award or with an F32 were mostly evenly spread across the institutional quartiles (Fig 3A and S5 Table). Over 35 percent of FTR01 awardees with a prior K01 were in first quartile institutions and this came at the expense of those in fourth quartile institutions (Fig 3A and S5 Table). Over 75 percent of FTR01 awardees with a prior K08 and 70 percent of FTR01 awardees with a prior K23 were in first and second quartile institutions, and those in the third and fourth quartile were well below the 25 percent mark (Fig 3A and S5 Table). FTR01 awardees with a prior K99 award were above the 25 percent mark in the first, second, and third quartiles with a significant drop off for the fourth quartile (Fig 3A and S5 Table). These data indicate K01, K08, K23, and K99 awardees typically segregate into first and sometimes second quartile institutions.

Some departments and institutions hire their trainees as faculty members [21, 22]. Therefore, there are two paths from a training award to a faculty position: First, institutions could choose to retain training awardees as faculty, and, second, institutions could hire a training awardee through an external candidate job search. To determine the degree of retention versus external hiring for each training award, I determined the percentage of training awardees that received their training award and their R01 at the same institution. Just over 20 percent of F32 and K99 awardees stayed at the same institution for their training award and their R01 (S1 Fig). Over 60 percent of K01 awardees, nearly 70 percent of K08 awardees and over 80 percent of K23 awardees received their first R01 at the same institution as their training award (S1 Fig). These data indicate institutions have a strong preference for retaining their K01, K08, and K23 awardees.

I then reevaluated how FTR01 awardees that were external hires segregated among quartiles. FTR01 awardees with F32 and K01 awards were underrepresented in the first quartile and overrepresented in the third and fourth quartiles (Fig 3B and S5 Table). The distribution of FTR01 awardees with a K23 was flatter but these awardees were still overrepresented in the first quartile (Fig 3B and S5 Table). And the percentile values for FTR01 awardees with a K08 or K99 changed, but not enough to affect the overall trend in the distribution of awardees (Fig 3B and S5 Table). These data indicate that, when examining only external hires, those with an F32 or K01 tend to segregate to the third and fourth quartile, those with a K08 or K23 segregate into the first quartile, and those with a K99 are the most evenly distributed with most likely to receive an R01 in the second or third quartile.

Keeping the quartiles as defined as above, I next examined whether training awardees from specific quartiles comprised a larger than expected share of the FTR01 awardee pool. Because it takes significant time for a training awardee to receive an R01, analyses of F32 awardees were restricted to 2000 to 2007, analyses of K01, K08, and K23 awardees were restricted to 2000 to 2010, and analyses of K99 awardees were restricted to 2007 to 2010 (Fig 2C and Table 2). For each training award, trainees from first and second quartile institutions made up 60 to 85 percent of those who progressed to an R01 (Fig 3C and S5 Table). These values were largely unaffected when analyzing only external faculty hires (Fig 3D and S5 Table). These data indicate that those receiving a training award at a first or second quartile institution comprise the clear majority of the FTR01 population who previously held a training award.

The overrepresentation of training award recipients from first and second quartile institutions in the FTR01 population could indicate these training awardees are better at securing faculty positions and R01s, or they could indicate that first and second quartile institutions simply receive more training awards. To differentiate between these possibilities, I determined the percentage of training awardees in each quartile to receive an R01. Nearly 20 percent of

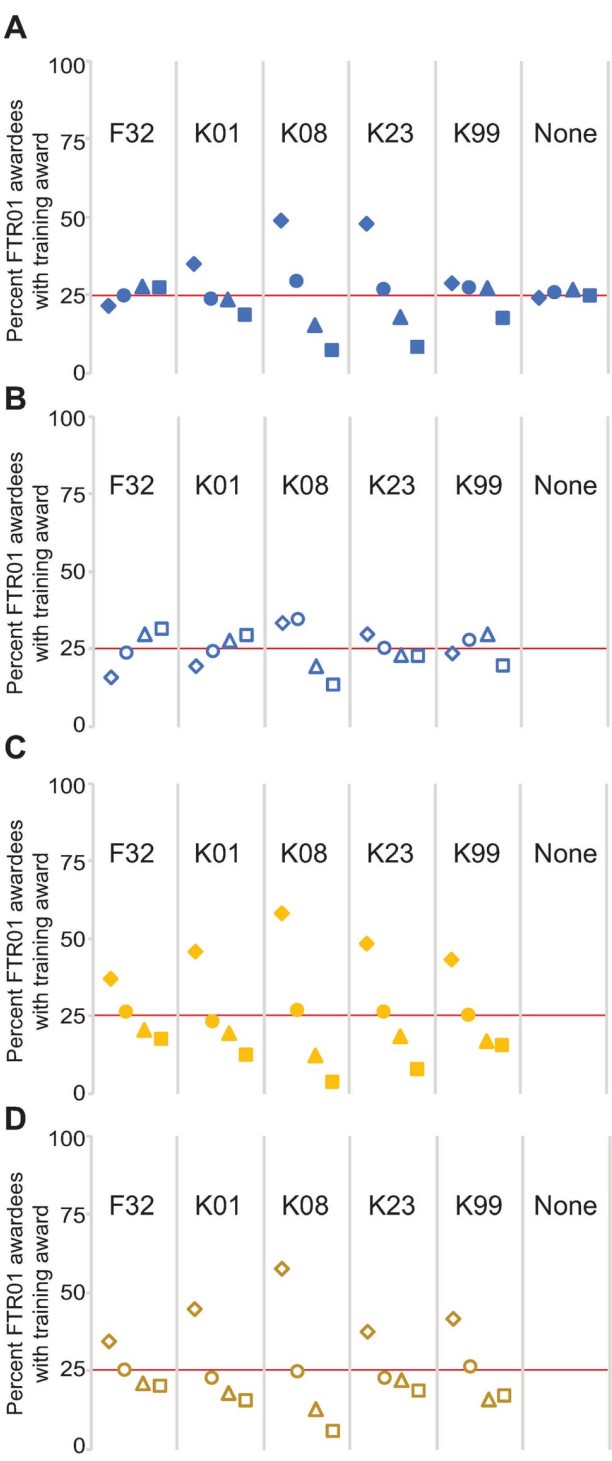

**Fig 3. Share of training awardees to eventually receive an R01 by institutional quartile.** Institutions were divided by quartile based on the number of FTR01s they received between 2000 and 2017, and the percentages of FTR01 awardees with the indicated training award were plotted. Quartiles are arranged from left to right with the first quartile as diamonds, the second quartile as circles, the third quartile as triangles, and the fourth quartile as squares. The red line indicates 25 percent. The data were evaluated based on the institutional quartile the FTR01 was received in for **(A)** all new faculty and **(B)** only external faculty hires or the institutional quartile the training award was received in for **(C)** all new faculty and **(D)** only external faculty hires.

those receiving an F32 between 2000 and 2007, 41 percent of K23 awardees between 2000 and 2010, and 57 percent of K99 awardees between 2007 and 2010 eventually received an R01, and all quartiles were within one standard deviation of the mean (S2A Fig and Table 2). About 42 to 44 percent of K01 and K08 awardees between 2000 and 2010 received an R01, but those receiving a K01 or K08 at a first quartile institution were more than one standard deviation away from the mean, while those in the fourth quartile for K01 awardees and the third and fourth quartiles for K08 awardees were more than one standard deviation below the mean (S2A Fig and Table 2). When analyzing only external faculty hires, nearly all values fell within one standard deviation of the mean (S2B Fig). These data indicate that, for those PIs that changed institutions between their training award and their first R01, the quartile in which they received a training award conferred minor advantages in eventually receiving an R01. Rather, the overrepresentation of first and second quartile training awardees in the FTR01 pool is likely due to those institutions receiving a larger number of training awards.

## The interaction of training award and R01 quartiles

To better understand the flow of researchers from training institutions to R01 institutions, I constructed 4X4 grids for each training award. The R01 quartile was arrayed along the horizontal axis and the training award quartile was arrayed along the vertical axis. The resulting grid indicates how training awardees from specific quartiles were distributed across quartiles for their R01.

For the entire population of FTR01 awardees, those that trained in a specific quartile were most likely to remain in that quartile (Fig 4A–4E). This effect was strongest for K01, K08, and K23 awardees (Fig 4B, 4C and 4D). Analyzing only external faculty hires revealed a different spread of FTR01 awardees across quartiles (Fig 4F–4J). Training awardees from first and second quartile institutions made up the majority of FTR01 awardees across all quartiles (Fig 4F–4J). In addition, a large proportion of third and fourth quartile F32 awardees received R01s at

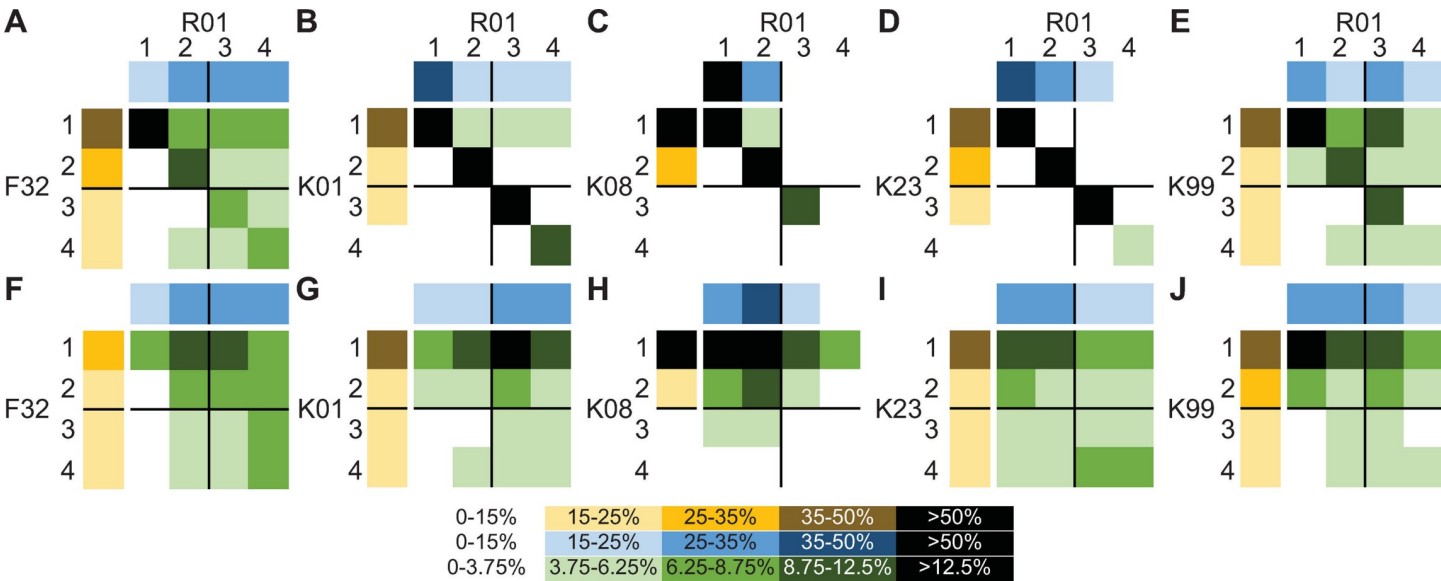

**Fig 4. Interactions of training award and R01 quartiles on the distribution of training awardees. (A)**-**(E)** All FTR01 awardees with the indicated training award and **(F)**-**(J)** FTR01 awardees who changed institutions between training award and R01. R01 quartiles are arrayed across the top (blue) and training award quartiles are arrayed down the left side (yellow). The distribution of FTR01 awardees based on training and R01 quartiles are in the 4X4 grid (green). The range of percentages signified by different colors is in the legend on the bottom. Percentages for the 4X4 grid are as for the quartiles divided by four (See Data collection and limitations).

fourth quartile institutions (Fig 4F). These data indicate that those who received a training award at a first quartile institution made up the largest fraction of faculty in institutions of all quartiles. Furthermore, the bottom left quadrant of each grid almost always has the smallest percentage of the population (Fig 4). This indicates that receiving a training award in a third and fourth quartile institution followed by an R01 at a first or second quartile institution is the least likely outcome for new faculty.

These data present a picture of how the aggregate FTR01 awardee pool from 2000 to 2017 is distributed across research institutions. To visualize how this distribution has changed over time, I plotted the composition of training awards in the FTR01 awardee pool in each quartile in 2000 and 2017. Those with no prior award in the 2000 FTR01 awardee pool ranged between 70 and 80 percent, and by 2017, this value was between 50 and 70 percent depending on quartile (Fig 5). Between 15 and 30 percent of FTR01 awardees previously held an F32, K08, or Other F/K award in 2000, and this value was 15 to 25 percent in 2017 (Fig 5). FTR01 awardees with a K01, K23, or K99 award were one percent or less of the 2000 FTR01 awardee pool, and they were 15 to 20 percent in 2017 (Fig 5). These data indicate that most of the increase in the proportion of FTR01 awardees with a prior NIH training award was due to the proliferation of K01, K23, and K99 awards. Furthermore, while there are differences in magnitude of changes across the quartiles, each quartile changed similarly in their composition of FTR01 awardees with prior NIH training awards.

## Data collection and limitations

Grant funding information from 1985 to 2017 was downloaded from NIH ExPORTER (https://exporter.nih.gov/). To begin my analyses, I isolated all R01 awardees from each fiscal year. Using the "Contact PI Person ID" unique identifier for each principal investigator, the list of R01 holders from each year was deduplicated. The deduplicated unique identifiers for each year were arrayed next to each other, and standard Excel (Microsoft Excel for Mac V15.33; Microsoft Corp., Redmond, WA, USA) formulas were used to determine whether a PI in a given year had ever previously received an R01 in a previous year stretching back to 1985. This was completed iteratively for each fiscal year. Those in each year that had never received a prior R01 were termed first-time R01 (FTR01) awardees.

I analyzed FTR01 awardees as opposed to the NIH-derived New Investigator (NI) or Early-Stage Investigator (ESI) pools. NIs and ESIs are investigators that have not "previously competed successfully as a [Program Director/Primary Investigator] for a substantial independent research award" [23]. However, what qualifies as a "substantial independent research award" can change based on the introduction or elimination of specific grant mechanisms. Furthermore, it is not readily apparent who is an ESI or NI with publicly accessible data, whereas FTR01 awardees are readily identifiable. For these reasons, I chose to analyze FTR01 awardees.

The unique identifiers of FTR01 awardees were compared to the unique identifiers of all F and K-series awardees from 1985 to 2017 using standard Excel formulas. The F and K-awardee list was limited to type 1/new awards. This matching method allowed the assignment of data to specific PIs including, but not limited to, the year and location a training award was received and the year and location an R01 was received. For this study, "training awards" refers to F and K-series awards collectively and does not include T-series training grants.

## Study time frame

Grant data through NIH ExPORTER is available from 1985 through the present. To analyze training awards received prior to the first R01, I started the analysis of FTR01 awardees in 2000 to give 15 years of prior award information.

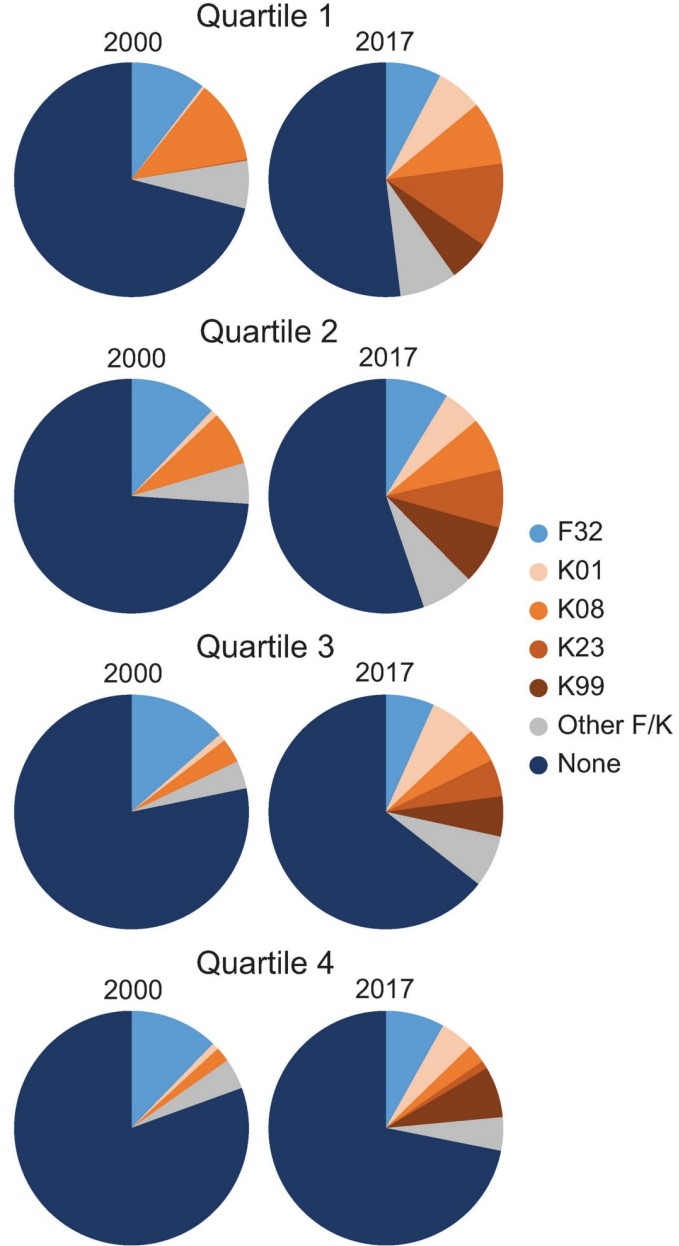

**Fig 5. Change in composition of the FTR01 population with specific training awards between 2000 and 2017 by quartile.**

Some training mechanisms are also available to faculty, and it is possible some of these training awards were received after the PI's R01. To ensure the analyses here examined only those PIs who received a training award prior to their R01, the year a PI received a training award was subtracted from the year of their first R01. Negative values indicated the PI received the training award after their first R01, and zeroes indicated the PI received the training award the same year as their first R01. There were no instances of negative values or zeroes for F32 or K99 awards. There were 30 negative values and zeroes for K01 awardees (of 1227 awards, 2.4 percent), 12 negative values and zeroes for K08 awardees (of 2656 awards, 0.5 percent), and 19

negative values and zeroes for K23 awardees (of 1114 awards, 1.7 percent). Training awardees who received their R01 the same year or after their first R01 were excluded from the analysis.

## Institutional analyses

Institutions are not always consistently named in NIH databases. For example, "University of California San Francisco" was used from 1985 through 2007 and "University of California, San Francisco" was used from 2008 through 2017. While these are clearly the same institution, the punctuation caused Excel to consider these as separate entities. Furthermore, some institutions changed names, merged, or were otherwise divided in the database. I cleaned the datasets based on my understanding of the relationships between institutions to remove as many of these false separate entries as possible (S6 Table) [9]. In addition, hospitals and medical schools are often associated with universities, and employees and researchers move freely among them; however, some hospitals and med schools are fragmented from the university in NIH databases. To get a more accurate picture of grants to institutions, I consolidated grants to hospitals and med schools under the name of the degree-granting university they were associated with, similar to the NSF's Survey of Graduate Student and Postdoctorates in Science and Engineering (https://www.nsf.gov/statistics/srvygradpostdoc/).

I tallied the FTR01 awards for each institution between 2000 and 2017. The list of institutions was then arrayed from top to bottom by how many FTR01s that institution received over the time frame. Institutions were divided into quartiles based on the number of FTR01 awards. The first quartile is the first 25 percent of all FTR01s, starting from the top of the list, and is comprised of 12 institutions. The second quartile continues from the 25 to the 50 percent mark (23 institutions), the third quartile from 50 to 75 percent (46 institutions) and the fourth quartile from 75 to 100 percent (897 institutions). See S4 Table for the full list of institutions and divisions into quartiles.

To generate the heat maps examining the interactions of training awardee and R01 quartiles, the training award quartile was aligned along the left and the R01 quartile along the top. Raw numbers of grantees were generated by standard Excel analyses of the dataset and percentages were determined. As each quartile was divided another four times to provide 16 possible bins, the percentile ranges for the 4X4 grid were derived from the ranges for the quartiles by dividing each endpoint by four. For example, the first bin for the quartiles is 0–15 percent. And 15 / 4 = 3.75. Therefore, the first bin for the grid is 0–3.75 percent, the second bin is 3.75 to 6.25 (25 / 4), and so on.

## Discussion

Understanding the contours of hypercompetition and the realities of the environment in which today's young scientists compete for funding, publications, and jobs is critical to develop policies that will improve the biomedical research enterprise [1]. The data presented here explore a persistent perception among young researchers that NIH funding is a prerequisite for attaining a faculty position. While not a prerequisite, a clear shift is underway that favors biomedical faculty candidates with at least one prior training award. Furthermore, examining the distribution of FTR01 awardees with training awards across institutions highlights unappreciated complexities with regard to success rates and physical transitions to faculty positions.

The percentage of the FTR01 population with a prior F or K award has steadily increased since 2000, and this tracks largely with an increase in the number of awards available to trainees. These awards also appear to confer a competitive advantage on R01 applicants—F32 awardees are nearly 6 percent more likely to receive an R01 than those who tried and failed to

receive an F32, while K01, K08 and K23 awardees applying for an R01 had a nearly 25 percent greater chance of being awarded the R01 relative to those that applied for those K awards and failed [12, 14]. Together, these data indicate that training awards are increasing in importance for attaining an R01. However, the degree of importance is not knowable without investigating the number of training awardees who applied for an R01 and failed relative to those who didn't have a training award and failed to secure an R01. These data are not available in the public databases used in the current study.

FTR01 awardees accounted for about 35 percent of the 2017 FTR01 awardee pool. However, this value may be misleading. Most F and K mechanisms are reserved for U.S. citizens and permanent residents. Immigrant scientists make up roughly two-thirds of the postdoc workforce and 15 to 25 percent of the FTR01 awardee pool is foreign born [12, 24]. If the number of FTR01 awardees in a given year is adjusted to omit those who were ineligible for a training award, roughly half of the remaining FTR01 awardees received a training award before their R01 in 2017. Furthermore, the percentage of training awards among the FTR01 awardee population is overrepresented relative to the percentage of training awards among the postdoc population, supporting previous results indicating training awards confer a competitive advantage in attempts to land a faculty position [12, 14].

The expansion of K01, K23, and K99 awards in the FTR01 awardee pool is approximately matched by the decline in those with no prior award. The use of these mechanisms is relatively new: the K01 program was expanded in the late 1990s, K23 awards were introduced in 1999, and K99 awards were introduced in 2007 (Table 1) [18]. It took roughly 10 years for K01 and K23 awards to become a relatively stable part of the FTR01 awardee population, and, if the same holds for K99 awards, then the mechanism has a couple of years before reaching steady state. Therefore, the increase in FTR01 awardees with a training award could be due to institutions preferring to hire faculty with prior training awards or due to the grant system still adjusting to the perturbation caused by introducing new training grant mechanisms.

Rather than being in opposition, these two models likely reinforce one another. Training awards are introduced into the research system in response to pressures on young scientists to distinguish themselves from their peers and to ensure healthy populations of specific classes of trainee. As more training awards are made, the number of faculty candidates with training awards increases. This changes a department's perception of a faculty candidate's possible accomplishments and feeds into the evaluation process. The pressure on candidates to attain a training award increases and feeds the demand for more training awards. This vicious circle is consistent with the data presented here and also manifests in anecdotal reports of departments supposedly only interviewing faculty candidates with a K99 award, for example.

This cycle extends beyond receiving an NIH training award as potential faculty candidates compete for high-quality publications and grant awards. Other NIH grants, like the R21, R03, and the now defunct R29 award, are small grants for discrete research projects that could serve a similar career-development purpose as the awards studied here. Grants from other federal sources, such as the National Science Foundation or Veterans Affairs, and nonprofit organizations, such as the Howard Hughes Medical Institute or the American Heart Association, provide many of the same career benefits as NIH awards, and it is not clear how many of the FTR01 awardees without a prior NIH award had an award from a non-NIH organization. The highly selective and prestigious nature of these awards may portend even higher rates of conversion to faculty than observed for NIH training awards. However, these programs are small relative to the NIH system so the overall trends discussed here would likely be unaffected by including these awards. Understanding how FTR01 awardees with these non-NIH training awards are skewed across quartiles would be illuminating for understanding institutional preferences for funding track records.

## Strategies for gaining grant funding

There are two basic profiles of training awards among FTR01 awardees: the F32/K99 profile and the K08 profile. For the F32/K99 profile, FTR01 awardees are roughly evenly distributed across quartiles, and about 20 percent of awardees receive a training award and R01 at the same institution. For the K08 profile, FTR01 awardees are most concentrated in first and second quartile institutions and 60 percent or more of awardees receive a training award and R01 at the same institution. The population of FTR01 awardees with a K01 or K23 award resembles the K08 profile when looking at all of the data, while the profile of external faculty hires more closely resembles the F32/K99 profile.

While the attributes of FTR01 awardees with an F32 or K99 are quite similar in this analysis, the biggest difference is in the percentage of awardees that eventually receive an R01. F32 awards are targeted to newer postdocs to launch their postdoc career with independent funding [16]. K99 awards were designed to support senior postdocs and could be converted to an R00 after transitioning to a faculty position [16]. While about 20 percent of F32 awardees eventually received an R01, roughly 90 percent of K99 awardees converted their K99 to an R00 and about 50 percent went on to receive an R01 [25, 26]. These data suggest these mechanisms are working as designed—F32 awards to support new postdocs with uncertain research futures and K99 awards to support those seeking independent research careers as faculty members.

The phenomenon that some faculty received a training award and R01 while at the same institution is consistent with previous findings that those who receive their first grants at elite institutions often persist at those institutions [22, 27]. That this effect is specific to K01, K08, and K23 awardees gives a glimpse into strategies for attaining faculty positions and launching labs. K01, K08, and K23 awards are the only awards in this analysis that are available to young faculty members (Table 1). Therefore, some fraction of K01, K08, and K23 awardees could potentially secure faculty positions without prior NIH funding and receive a K01, K08, or K23 award as faculty before receiving an R01. In this instance, the interpretation of Fig 3A would be that faculty at first and second quartile institutions were more successful at obtaining a K award followed by an R01 than their counterparts at third and fourth quartile institutions. In this instance, the percentage of FTR01 awardees with an NIH award prior to attaining a faculty position would be lower than is presented here.

The data presented here add to the discussion of the skewed distribution of funding in the research enterprise and the limited hierarchy of institutions providing new faculty members [21, 28, 29]. The first quartile in the present analysis, accounting for 25 percent of FTR01s between 2000 and 2017, was comprised of 12 institutions. These same institutions accounted or 35 percent of F32 awardees, 45 percent of K01 awardees, 58 percent of K08 awardees, 37 percent of K23 awardees and 41 percent of K99 awardees to eventually receive an R01 after changing institutions (S4 Table). However, the quartile in which a training award was received did not affect the likelihood of receiving an R01. This indicates a bias across the research enterprise toward renewing the faculty ranks from a select number of universities. The data here cannot determine whether this bias is due to decisions made by institutions during the hiring process, NIH study sections during grant application review, or a mix of the two. Regardless, this bias impedes the research enterprise from becoming more diverse and inclusive while artificially limiting the avenues of research being pursued.

The data here should be interpreted with some caution. First, the present analysis is affected by a selection bias as it examines only those who successfully received an R01. These data do not capture the entire universe of new faculty members or R01 applicants, with or without training awards. Understanding how many young faculty members submitted R01 applications yet never received an R01, along with their training award track record and institutional

quartile, would give important context to this discussion. Second, the quartiles group institutions by average size of their FTR01 awardee classes, and they should not be confused with assumptions of institutional prestige. Some prestigious institutions were outside of the first quartile—Stanford University, for example, was in the second quartile and Princeton University was in the fourth quartile. Therefore, the quartile rankings here are not necessarily aligned with the perception of institutional prestige.

The data presented here demonstrate some of the power of public databases for understanding the dynamics of the biomedical research enterprise. NIH ExPORTER contains a wealth of information, and linking this database with other databases could increase the power of the general community's ability to analyze the biomedical research enterprise. Furthermore, the NIH has a variety of datasets it is prohibited from making public but that are available to researchers on request (for example, see [12, 14]). Overlaying publicly available information, like that presented here, with non-public information, like the numbers and types of grant applications submitted by trainees and faculty, would be insightful. For example, do F32 awardees have to submit more or fewer applications before receiving an R01 than K awardees? And have the application and funding dynamics for those without training awards changed substantially over time? Beyond this, significant work is needed to better understand the multifactorial interactions among a faculty member's training award track record, their publication history, and their training location and how they play into where they land a faculty position and how this affects R01 applications and awards. Furthermore, understanding how these forces shape hypercompetition is needed to craft policies that effectively relieve these pressures.

## Supporting information

**S1 Fig. The percentage of training awardees who remained at their training institution for their R01.**
(EPS)

**S2 Fig. Likelihood of training awardees to eventually receive an R01 by institutional quartile. (A)** The percentage of all training awardees within each quartile that eventually received an R01 was plotted. Quartiles as in Fig 3. Solid lines indicate the average percentage of those with the indicated training award to receive an R01 between 2000 and 2007 (F32), 2000 and 2010 (K01, K08, and K23) and 2007 and 2010 (K99). Dashed lines indicate one standard deviation from the mean. **(B)** As in (A) but for only external faculty hires.
(EPS)

**S1 Table. Percentage of first-time R01 awardees with prior training awards.**
(XLSX)

**S2 Table. Award mechanisms in each category.**
(XLSX)

**S3 Table. Percentage of FTR01 awardees with indicated training awards.**
(XLSX)

**S4 Table. List of institutions arrayed from most to least FTR01s from 2000 to 2017.**
(XLSX)

**S5 Table. Percent values for datapoints in indicated figures.**
(XLSX)

**S6 Table. Institutional names found in NIH RePORTer that were aggregated in the current study.**
(XLSX)

## Acknowledgments

I thank Katherine Pickett, Melissa Vaught, George Santangelo, Gary McDowell, Judith Kimble, Needhi Bhalla, and Bruce Alberts for helpful comments on an earlier version of this manuscript.

## Author Contributions

**Conceptualization:** Christopher L. Pickett.

**Data curation:** Christopher L. Pickett.

**Formal analysis:** Christopher L. Pickett.

**Funding acquisition:** Christopher L. Pickett.

**Investigation:** Christopher L. Pickett.

**Methodology:** Christopher L. Pickett.

**Project administration:** Christopher L. Pickett.

**Resources:** Christopher L. Pickett.

**Software:** Christopher L. Pickett.

**Validation:** Christopher L. Pickett.

**Visualization:** Christopher L. Pickett.

**Writing – original draft:** Christopher L. Pickett.

**Writing – review & editing:** Christopher L. Pickett.

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
