## [Decision Letter · Decision Letter 0]

27 Aug 2019

[EXSCINDED]

PONE-D-19-22271

The Increasing Importance of Fellowships and Career Development Awards in the Careers of Early-Stage Biomedical Academic Researchers

PLOS ONE

Dear Dr. Pickett,

Thank you for submitting your manuscript to PLOS ONE. After careful consideration, we feel that it has merit but does not fully meet PLOS ONE’s publication criteria as it currently stands. Therefore, we invite you to submit a revised version of the manuscript that addresses the points raised during the review process.

Specifically: The  authors should consider grant awards specific to early stages of biomedical academic investigators  in data analysis and interpretations appropriately. 

We would appreciate receiving your revised manuscript by Oct 11 2019 11:59PM. To enhance the reproducibility of your results, we recommend that if applicable you deposit your laboratory protocols in protocols.io, where a protocol can be assigned its own identifier (DOI) such that it can be cited independently in the future. For instructions see: http://journals.plos.org/plosone/s/submission-guidelines#loc-laboratory-protocols

We look forward to receiving your revised manuscript.

Kind regards,

Dr. Sakamuri V. Reddy

Academic Editor

PLOS ONE

Journal Requirements:

Reviewers' comments:

Reviewer's Responses to Questions

**Comments to the Author**

1. Is the manuscript technically sound, and do the data support the conclusions?

Reviewer #1: Yes

Reviewer #2: Partly

2. Has the statistical analysis been performed appropriately and rigorously? 

Reviewer #1: N/A

Reviewer #2: Yes

3. Have the authors made all data underlying the findings in their manuscript fully available?

Reviewer #1: Yes

Reviewer #2: Yes

4. Is the manuscript presented in an intelligible fashion and written in standard English?

Reviewer #1: Yes

Reviewer #2: Yes

5. Review Comments to the Author

Reviewer #1: In this manuscript Dr. Pickett used the National Institutes of Health (NIH) award database to investigate the role of F31 and F32 pre- and post-doctoral training awards and NIH career development awards (K awards) in the success of an investigator in securing an R01 award for the first time (FTR01). He found that the proportion of FTR01 awardees without a previous F or K award has been steadily declining since 2000, while the percentage of those with a previous K award has been steadily increasing in that time period. In particular, a high proportion of those receiving a K99 award seem to go on to obtain an R01 given sufficient time (i.e., about ten years so that those receiving their K99s between 2007 and 2012 have been quite successful to date at becoming FTR01 recipients). FTR01 awardees who had been able to obtain previous K awards in “first-tier” institutions, which were defined as having the greatest number of FTR01 awardees, were in general over-represented among new faculty hires in first-tier institutions. All institutions showed a decrease in the number of FTR01 recipients without a prior F or K award; the first- and second-tier institutions showed the lowest percentage of these FTR01 awardees without a previous award, mostly due to the fact that a larger proportion of their FTR01 recipients had received K awards. These data are interesting and needed for an informed discussion of NIH funding policies. There are a few relatively minor issues that require attention.

Major Points:

(1) F30 awards are also training grants. Were they such a small percentage that they were not considered?

(2) The NIH R29, although technically an R grant was geared to first-time NIH applicants and so for many played the role of a career development award. It is not clear whether or not the author included these R29 awards in his analysis but perhaps should.

Minor points:

(1) For Figure 3 (and supplemental Figure 2) it is suggested that the first-, second-, third- and fourth-tier institutions be represented by different symbols or colors.

(2) The author might wish to remind the reader to whom each K award is geared (MD versus PhD) and perhaps discuss whether there are differences related to degree.

(3) The Medical College of Georgia has undergone several name changes since 2000, first to Georgia Health Sciences University, then to Georgia Regents University and finally to Augusta University; the data from these “different” institutions should be combined.

(4) The author mentions the possibility of FTR01 awardees without prior F or K awards having received instead a career development award from an alternative agency on the third page of the Discussion. It is felt that this idea deserves a more extensive discussion, especially since certain organizations, such as the American Heart Association, have career development awards that jump-start a number of careers, particularly in cardiovascular research. Also, what role, if any, do career development awards from the Department of Veterans Affairs play in the ability of investigators to obtain R01 awards?

(5) On the first page of the results, it should be “principal investigator” rather than “principle investigator”.

Reviewer #2: This paper by Pickett describes an analysis of first-time NIH R01 grantees. He reports that an increasing percentage of new R01 winners has received a prior NIH F or K training grant. Furthermore, he follows the “flow” of researchers from training grants to independent appointments, and he finds that “top” institutions tend to produce the most new investigators. Overall, I think that most of the analysis is sound and informative. I have two major issues with the manuscript, and a few minor ones, described below:

1) Pickett divides institutions into quartiles based on the number of first-time R01s produced each year. I would anticipate that this would divide institutions based on size, rather than based on quality or prestige or research output or anything else. Indeed, looking at the makeup of each quartile, you can find MIT, Rockefeller, and the Salk in the bottom quartile, along with East Carolina University. The University of Pittsburgh is in the top quartile. (Pickett notes the same issue, when he mentions how Stanford is in the 2nd quartile and Princeton in the 4th). This reinforces my belief that he has divided institutions based on their total size (or, more accurately, the number of new faculty hired each year). Yet, in the abstract, he states that this data shows that “training awardees from top institutions were overrepresented in the faculty of the majority of institutions”. I do not think that that is an acceptable interpretation of the data.

I think that the question of researcher “flow” between institutions is important, but this needs to either be explained or analyzed differently. The approach requiring the least effort would probably be to simply change the wording in the paper: remove words like “top” when describing this division, and instead state clearly that this is likely a reflection of institutional size. That would also mean moderating the conclusions somewhat. Alternately, Pickett could choose to re-analyze the data (or add a supplemental analysis) that uses a more accurate division based on institutional prestige: US News Rankings? Nature’s research productivity index? Percent of faculty in the national academy? With a division based on one of those criteria, then I think that Pickett can reasonably refer to “top” institutions (with the caveat that of course no metric is perfect).

2) A somewhat similar issue (though the manuscript does a better job of directly addressing it): the title of the paper is “The Increasing Importance of Fellowships and Career Development Awards in the Careers of Early-Stage Biomedical Academic Researchers.” Yet, the data shows that the increasing percent of training awardees among new r01 recipients largely tracks with the increasing number of training awards, particularly the K99. If 20% of new faculty and 25% of new r01 recipients had training awards in 1999, while in 2017 it’s 30% and 35%, can you really say that fellowships are “increasingly important”? The lack of a “denominator” for these comparisons makes drawing such a conclusion difficult. I think that this can be dealt with by changing the title and the language in a few places – maybe something like “An increasing number of new NIH investigators have previously received an NIH training grant” – to more-accurately reflect your findings.

Minor points:

1) “R01s are of sufficient size and duration to sustain an independent research program” – not accurate.

2) “These data indicate institutions have a strong preference for retaining their K01, K08, and K23 awardees.” – These K grants can be awarded to early-career faculty, right? Is it possible that the apparent retention of K grantees can be explained because they were actually awarded at institutions where these individuals are currently assistant professors?

3) “These data suggest first and sometimes second quartile institutions prefer faculty candidates with K01, K08, K23, and K99 awards.” – can you draw this conclusion without knowing how many awardees applied to these institutions?

6. PLOS authors have the option to publish the peer review history of their article (what does this mean?). If published, this will include your full peer review and any attached files.

Reviewer #1: No

Reviewer #2: No

---

## [Author Response · Author response to Decision Letter 0]

13 Sep 2019

(Please note the page number references are to the revised version of the manuscript with the figures interwoven, not the tracked changes version requested by PLOS ONE.)

Reviewer 1

• F30 awards are also training grants. Were they such a small percentage that they were not considered?

I collected the FTR01 awardees and asked what F and K-series awards these awardees held. Nine F-series fellowships were represented in this cohort, including the F30 MD/PhD predoctoral fellowship. Of all of the F-series fellowships held by FTR01 awardees, 0.98% were F30 awards. For reference, 81.2% of F awards were F32 awards and 15.8% were F31 awards.

As the reviewer notes, F30 awards are important awards to trainees, even if they are not highly represented in the FTR01 awardee pool. I have included a new supplemental table, Table S2, that breaks down the percentage of all F and K-series awards in the FTR01 awardee pool

• The NIH R29, although technically an R grant was geared to first-time NIH applicants and so for many played the role of a career development award. It is not clear whether or not the author included these R29 awards in his analysis but perhaps should.

The question addressed in this manuscript is whether FTR01 awardees held prior F or K awards. The reviewer is asking a slightly different question—do FTR01 awardees receive any NIH award that would provide career development opportunities prior their first R01? This analysis would include those receiving R29 awards, which were designed to be a young faculty member’s first major research award. The R29 is no longer supported by the NIH, but R03 and R21 awards fill a similar niche in that they are small awards for a short amount of time that may be attained before an R01.

I focused on F and K awards because they can be secured by trainees, whereas R-series awards are almost always awarded to those with faculty, or similar level, appointments. Because I was interested in not just whether FTR01 awardees had prior training awards, but how those awards affected their movement to faculty positions, I only analyzed awards that could be won by trainees. Therefore, I believe an analysis of R29 awards to be beyond the scope of the current investigation.

I think the reviewer raises an important question, though, and I included text in the Discussion on p. 29 to indicate other awards could be used in similar ways to the F and K awards I analyze here.

• For Figure 3 (and supplemental Figure 2) it is suggested that the first-, second-, third- and fourth-tier institutions be represented by different symbols or colors.

I made the suggested changes so that all quartiles are different symbols.

• The author might wish to remind the reader to whom each K award is geared (MD versus PhD) and perhaps discuss whether there are differences related to degree.

I appreciate this comment. Table 1 explains the MD vs. PhD information, but I only make reference to this table at the beginning of the Results section. More references to the table throughout the document would address the reviewers concern, and I have included several more references to Table 1 in the Results and the Discussion.

• The Medical College of Georgia has undergone several name changes since 2000, first to Georgia Health Sciences University, then to Georgia Regents University and finally to Augusta University; the data from these “different” institutions should be combined.

I thank the reviewer for this information. I had combined Georgia Health Sciences University, Georgia Regents University and Augusta University, but I had not included the Medical College of Georgia. I have rectified this and updated Table S6 accordingly.

• The author mentions the possibility of FTR01 awardees without prior F or K awards having received instead a career development award from an alternative agency on the third page of the Discussion. It is felt that this idea deserves a more extensive discussion, especially since certain organizations, such as the American Heart Association, have career development awards that jump-start a number of careers, particularly in cardiovascular research. Also, what role, if any, do career development awards from the Department of Veterans Affairs play in the ability of investigators to obtain R01 awards?

I have expanded on a paragraph in the Discussion on p. 29 to address the role non-NIH awards may have in hiring and R01 funding decisions. 

• On the first page of the results, it should be “principal investigator” rather than “principle investigator”.

These changes have been made.

Reviewer 2:

• Pickett … states that this data shows that “training awardees from top institutions were overrepresented in the faculty of the majority of institutions”. I do not think that that is an acceptable interpretation of the data.

I think that the question of researcher “flow” between institutions is important, but this needs to either be explained or analyzed differently. The approach requiring the least effort would probably be to simply change the wording in the paper: remove words like “top” when describing this division, and instead state clearly that this is likely a reflection of institutional size. That would also mean moderating the conclusions somewhat. Alternately, Pickett could choose to re-analyze the data (or add a supplemental analysis) that uses a more accurate division based on institutional prestige: US News Rankings? Nature’s research productivity index? Percent of faculty in the national academy? With a division based on one of those criteria, then I think that Pickett can reasonably refer to “top” institutions (with the caveat that of course no metric is perfect).

This is an excellent point and I appreciate the reviewer bringing it up. I was trying to avoid discussions of prestige because it is highly subjective. Apparently, I was not entirely successful in editing out this language.

As the reviewer points out, the ranking of institutions into quartiles by average size of their annual FTR01 class likely reflects the size of institutions rather than the prestige of the institution. I have revised the manuscript to remove descriptors such as “top” to describe institutions, and, when appropriate, to discuss these institutions by quartile rather than other subjective qualifiers.

• A somewhat similar issue (though the manuscript does a better job of directly addressing it): the title of the paper is “The Increasing Importance of Fellowships and Career Development Awards in the Careers of Early-Stage Biomedical Academic Researchers.” Yet, the data shows that the increasing percent of training awardees among new R01 recipients largely tracks with the increasing number of training awards, particularly the K99. If 20% of new faculty and 25% of new R01 recipients had training awards in 1999, while in 2017 it’s 30% and 35%, can you really say that fellowships are “increasingly important”? The lack of a “denominator” for these comparisons makes drawing such a conclusion difficult. I think that this can be dealt with by changing the title and the language in a few places – maybe something like “An increasing number of new NIH investigators have previously received an NIH training grant” – to more-accurately reflect your findings.

The point I tried to make regarding “increasing importance” combines the data presented in the manuscript with previous work done by others. The reviewer is correct that the data presented here show that the number of FTR01 awardees with prior training awards tracks with an increase in the overall number of training awards. In addition, Heggeness et al (ref 12 in the manuscript) showed that people with an F32 applying for an R01 had a 6% greater chance of being awarded an R01 relative to those that applied for an F32 and did not receive it. Similarly, Nikaj and Lund (ref 14 in the manuscript) demonstrated that K01, K08 and K23 awardees applying for an R01 had a nearly 25% greater chance of being awarded the R01 relative to those that applied for those K awards and failed.

The increasing number of training awards across the timeframe analyzed combined with the competitive advantage they confer led me to conclude the awards were increasing in importance. However, the reviewer is correct that until we know how many people with training awards applied for an R01 and failed relative to those who didn’t have a training award and failed to secure an R01, I am not able quantify the degree of importance of training awards to success in the R01 competition.

Because the data are consistent with increasing importance, I have let this language stand. I have written a new paragraph in the Discussion on p. 27 to explicitly address how I arrived at this conclusion along with noting the caveat brought up by the reviewer. 

• “R01s are of sufficient size and duration to sustain an independent research program” – not accurate.

I have removed the offending phrase. The sentence on p. 4 now reads, “R01s are highly sought by early-career faculty and support more early-career faculty than any other NIH grant mechanism.”

• “These data indicate institutions have a strong preference for retaining their K01, K08, and K23 awardees.” – These K grants can be awarded to early-career faculty, right? Is it possible that the apparent retention of K grantees can be explained because they were actually awarded at institutions where these individuals are currently assistant professors?

This is a good point, and I have expanded the discussion of this phenomenon on p. 30 of the Discussion.

• “These data suggest first and sometimes second quartile institutions prefer faculty candidates with K01, K08, K23, and K99 awards.” – can you draw this conclusion without knowing how many awardees applied to these institutions?

This is a good point and I appreciate the reviewer bringing it up. I have modified the text to read “These data indicate K01, K08, K23, and K99 awardees typically segregate into first and sometimes second quartile institutions.” on p. 13.

---

## [Decision Letter · Decision Letter 1]

27 Sep 2019

PONE-D-19-22271R1

The increasing importance of fellowships and career development awards in the careers of early-stage biomedical academic researchers

PLOS ONE

Dear Dr. Pickett,

Thank you for submitting your manuscript to PLOS ONE. After careful consideration, we feel that it has merit but does not fully meet PLOS ONE’s publication criteria as it currently stands. Therefore, we invite you to submit a revised version of the manuscript that addresses the points raised during the review process.

ACADEMIC EDITOR: Please revise the manuscript with changes suggested by the expert reviewer-1 as noted below.

We would appreciate receiving your revised manuscript by Nov 11 2019 11:59PM. To enhance the reproducibility of your results, we recommend that if applicable you deposit your laboratory protocols in protocols.io, where a protocol can be assigned its own identifier (DOI) such that it can be cited independently in the future. For instructions see: http://journals.plos.org/plosone/s/submission-guidelines#loc-laboratory-protocols

We look forward to receiving your revised manuscript.

Kind regards,

Dr. Sakamuri V. Reddy

Academic Editor

PLOS ONE

Reviewers' comments:

Reviewer's Responses to Questions

**Comments to the Author**

1. If the authors have adequately addressed your comments raised in a previous round of review and you feel that this manuscript is now acceptable for publication, you may indicate that here to bypass the “Comments to the Author” section, enter your conflict of interest statement in the “Confidential to Editor” section, and submit your "Accept" recommendation.

Reviewer #1: (No Response)

Reviewer #2: All comments have been addressed

2. Is the manuscript technically sound, and do the data support the conclusions?

Reviewer #1: Yes

Reviewer #2: Yes

3. Has the statistical analysis been performed appropriately and rigorously? 

Reviewer #1: N/A

Reviewer #2: Yes

4. Have the authors made all data underlying the findings in their manuscript fully available?

Reviewer #1: Yes

Reviewer #2: Yes

5. Is the manuscript presented in an intelligible fashion and written in standard English?

Reviewer #1: Yes

Reviewer #2: Yes

6. Review Comments to the Author

Reviewer #1: The author has addressed previous critiques well and only one minor issue remains for this reviewer. The author indicates that "F32 awardees are nearly 6 percent more likely to receive an R01 than those who tried and failed to receive an F32...." as well as similar statements for other award mechanisms. However, the data available in NIH ExPORTER are only those applications that are funded and there is no indication in these data of unsuccessful applications for grant awards. Therefore, it is not obvious how this conclusion is reached. If others have performed such a study, a citation must be provided. Otherwise, just because a first-time R01 awardee did not have a previous F32 (or other training award) does not mean that he or she tried and failed as there are other mechanisms of support and perhaps the awardee never even applied for an F32 (or other training award).

Reviewer #2: I believe that the author has addressed my concerns. I think that the article is now fit for publication.

7. PLOS authors have the option to publish the peer review history of their article (what does this mean?). If published, this will include your full peer review and any attached files.

Reviewer #1: No

Reviewer #2: No

---

## [Author Response · Author response to Decision Letter 1]

30 Sep 2019

Reviewer 1

• The author has addressed previous critiques well and only one minor issue remains for this reviewer. The author indicates that "F32 awardees are nearly 6 percent more likely to receive an R01 than those who tried and failed to receive an F32...." as well as similar statements for other award mechanisms. However, the data available in NIH ExPORTER are only those applications that are funded and there is no indication in these data of unsuccessful applications for grant awards. Therefore, it is not obvious how this conclusion is reached. If others have performed such a study, a citation must be provided. Otherwise, just because a first-time R01 awardee did not have a previous F32 (or other training award) does not mean that he or she tried and failed as there are other mechanisms of support and perhaps the awardee never even applied for an F32 (or other training award).

The data quoted in the indicated sentence are from previously published papers—Heggeness et al and Nikaj and Lund, references 12 and 14 respectively. I forgot to include these references in my revision, and I thank the reviewer for pointing this out. I have rectified this issue.

---

## [Editor Report · Decision Letter 2]

2 Oct 2019

The increasing importance of fellowships and career development awards in the careers of early-stage biomedical academic researchers

PONE-D-19-22271R2

Dear Dr. Pickett,

We are pleased to inform you that your manuscript has been judged scientifically suitable for publication and will be formally accepted for publication once it complies with all outstanding technical requirements.

With kind regards,

Dr. Sakamuri V. Reddy

Academic Editor

PLOS ONE
---

## [Editor Report · Acceptance letter]

8 Oct 2019

PONE-D-19-22271R2 

The increasing importance of fellowships and career development awards in the careers of early-stage biomedical academic researchers 

Dear Dr. Pickett:

I am pleased to inform you that your manuscript has been deemed suitable for publication in PLOS ONE. Congratulations! Your manuscript is now with our production department. 

With kind regards,

on behalf of

Dr. Sakamuri V. Reddy 

Academic Editor

PLOS ONE